# Neighborhood Gradient Clustering: An Efficient Decentralized Learning Method for Non-IID Data Distributions

## Abstract

Decentralized learning algorithms enable the training of deep learning models over large distributed datasets generated at different devices and locations, without the need for a central server. In practical scenarios, the distributed datasets can have significantly different data distributions across the agents. The current state-of-the-art decentralized algorithms mostly assume the data distributions to be Independent and Identically Distributed (IID). This paper focuses on improving decentralized learning over non-IID data distributions with minimal compute and memory overheads. We propose *Neighborhood Gradient Clustering (NGC)*, a novel decentralized learning algorithm that modifies the local gradients of each agent using self- and cross-gradient information. Cross-gradients for a pair of neighboring agents are the derivatives of the model parameters of an agent with respect to the dataset of the other agent. In particular, the proposed method replaces the local gradients of the model with the weighted mean of the self-gradients, model-variant cross-gradients (derivatives of the received neighbors' model parameters with respect to the local dataset - computed locally), and data-variant cross-gradients (derivatives of the local model with respect to its neighbors' datasets - received through communication). The data-variant cross-gradients are aggregated through an additional communication round without breaking the privacy constraints of the decentralized setting. Further, we present *CompNGC*, a compressed version of *NGC* that reduces the communication overhead by $32\times$ through cross-gradient compression. We demonstrate the efficiency of the proposed technique over non-IID data sampled from various vision and language datasets trained on diverse model architectures, graph sizes and topologies. Our experiments demonstrate that *NGC* and *CompNGC* either remain competitive or outperform (by $0-6\%$) the existing state-of-the-art (SoTA) decentralized learning algorithm over non-IID data with significantly less compute and memory requirements. Further, our experiments show that the model-variant cross-gradients information available locally at each agent can improve the performance over non-IID data by $1-35\%$ without any additional communication cost.

## 1 Introduction

The remarkable success of deep learning is mainly attributed to the availability of humongous amounts of data and compute power. Large amounts of data is generated on a daily basis at different devices all over the world which could be used to train powerful deep learning models. Collecting such data for centralized processing is not practical because of the communication and privacy constraints. To address this concern, a new interest in developing distributed learning algorithms Agarwal & Duchi (2011) has emerged. Federated learning (centralized learning) Konečný et al. (2016) is a popular setting in the distributed machine learning paradigm, where the training data is kept locally at the edge devices and a global shared model is learnt by aggregating the locally computed updates through a coordinating central server. Such a setup requires continuous communication with a central server which becomes a potential bottleneck Haghighat et al. (2020). This has motivated the advancements in decentralized machine learning.

Decentralized machine learning is a branch of distributed learning which focuses on learning from data distributed across multiple agents/devices. Unlike Federated learning, these algorithms assume that the agents are connected peer to peer without a central server. It has been demonstrated that decentralized learning algorithms Lian et al. (2017) can perform comparable to centralized algorithms on benchmark vision datasets. Lian et al. (2017) present Decentralised Parallel Stochastic Gradient Descent (D-PSGD) by combining SGD with gossip averaging algorithm Xiao & Boyd (2004). Further, the authors analytically show that the convergence rate of D-PSGD is similar to its centralized counterpart Dean et al. (2012). Balu et al. (2021) propose and analyze Decentralized Momentum Stochastic Gradient Descent (DMSGD) which introduces momentum to D-PSGD. Assran et al. (2019) introduce Stochastic Gradient Push (SGP) which extends D-PSGD to directed and time varying graphs. Tang et al. (2019); Koloskova et al. (2019) explore error-compensated compression techniques (Deep-Squeeze and CHOCO-SGD) to reduce the communication cost of P-DSGD significantly while achieving same convergence rate as centralized algorithms. Aketi et al. (2021) combined Deep-Squeeze with SGP to propose communication efficient decentralized learning over time-varying and directed graphs. Recently, Koloskova et al. (2020) proposed a unified framework for the analysis of gossip based decentralized SGD methods and provide the best known convergence guarantees.

The key assumption to achieve state-of-the-art performance by all the above mentioned decentralized algorithms is that the data is independent and identically distributed (IID) across the agents. In particular, the data is assumed to be distributed in a uniform and random manner across the agents. This assumption does not hold in most of the real-world applications as the data distributions across the agent are significantly different (non-IID) based on the user pool Hsieh et al. (2020). The effect of non-IID data in a peer-to-peer decentralized setup is a relatively under-studied problem. There are only a few works that try to bridge the performance gap between IID and non-IID data distributions for a decentralized setup. Note that, we mainly focus on a common type of non-IID data, widely used in prior works Tang et al. (2018); Hsieh et al. (2020); Lin et al. (2021); Esfandiari et al. (2021): a skewed distribution of data labels across agents. Tang et al. (2018) proposed $D^2$ algorithm that extends D-PSGD to non-IID data distribution. However, the algorithm was demonstrated on only a basic LENET model and is not scalable to deeper models with normalization layers. Lin et al. (2021) replace local momentum with Quasi-Global Momentum (QGM) and improve the test performance by $1-20\%$. But the improvement in accuracy is only $1-2\%$ in case of highly skewed data distribution as shown in Aketi et al. (2022). Most recently, Esfandiari et al. (2021) proposed Cross-Gradient Aggregation (CGA) and a compressed version of CGA (CompCGA), claiming state-of-the-art performance for decentralized learning algorithms over completely non-IID data. CGA aggregates *cross-gradient* information, i.e., derivatives of its model with respect to its neighbors' datasets through an additional communication round. It then updates the model using projected gradients based on quadratic programming. The cross-gradient and self-gradient terms are formally defined in Section 3. CGA and CompCGA require a very slow quadratic programming step Goldfarb & Idnani (1983) after every iteration for gradient projection which is both compute and memory intensive. This work focuses on the following question: *Can we improve the performance of decentralized learning over non-IID data with minimal compute and memory overhead?*

In this paper, we propose *Neighborhood Gradient Clustering* (*NGC*) to handle non-IID data distributions in peer-to-peer decentralized learning setups. Firstly, we classify the gradients available at each agent into three types, namely self-gradients, model-variant cross-gradients, and data-variant cross-gradients (see Section 3). The self-gradients (or local gradients) are the derivatives computed at each agent on its model parameters with respect to the local dataset. The model-variant cross-gradients are the derivatives of the received neighbors' model parameters with respect to the local dataset. These gradients are computed locally at each agent after receiving the neighbors' model parameters. Communicating the neighbors' model parameters is a necessary step in any gossip based decentralized algorithm Lian et al. (2017). The data-variant cross-gradients are the derivatives of the local model with respect to its neighbors' datasets. These gradients are obtained through an additional round of communication. We then cluster the gradients into a) *model-variant cluster* with self-gradients and model-variant cross-gradients, and b) *data-variant cluster* with self-gradients and data-variant cross-gradients. Finally, the local gradients are replaced with the weighted average of the cluster means. The main motivation behind this modification is to account for the high variation in the computed local gradients (and in turn the model parameters) across the neighbors due to the non-IID nature of the data distribution.

The proposed technique has two rounds of communication at every iteration to send model parameters and data-variant cross-gradients which incurs $2\times$ communication cost compared to traditional decentralized algorithms (D-PSGD). To reduce the communication overhead, we propose compressed version of *NGC* (*CompNGC*) by compressing the additional round of cross-gradient communication. Moreover, if the weight associated with data-variant cluster is set to 0 then *NGC* does not require an additional round of communication. We validate the performance of proposed algorithm on CIFAR-10 dataset over various model architectures and graph topologies. We compare the proposed algorithm with D-PSGD, CGA, and CompCGA and show that we can achieve superior performance over non-IID data compared to the current state-of-the-art approach. We also report the order of communication, memory and compute overheads required for *NGC* and CGA as compared to D-PSGD.

**Contributions:** In summary, we make the following contributions.

- We propose Neighborhood Gradient Clustering (*NGC*) for decentralized learning setting that utilizes self-gradients, model-variant cross-gradients and data-variant cross-gradients to improve the learning over non-IID data distribution (label-wise skew) among agents.

- We present compressed version of Neighborhood Gradient Clustering (*CompNGC*) that reduces the additional round of cross-gradients communication by $32\times$.

- Our experiments show that the proposed method either outperforms by $0-6\%$ or remains competitive with significantly less compute and memory requirements compared to current state-of-the-art decentralized learning algorithm over non-IID data at iso-communication cost.

- We also show that when the weight associated with data-variant cross-gradients is set to 0, *NGC* performs $1-35\%$ better than D-PSGD without any communication overhead by utilizing locally available model-variant cross-gradients information.

## 2 BACKGROUND

In this section, we provide the background on decentralized learning algorithm with peer-to-peer connections.

The main goal of the decentralized machine learning is to learn a global model using the knowledge extracted from the locally generated and stored data samples across $n$ edge devices/agents while maintaining the privacy constraints. In particular, we solve the optimization problem of minimizing global loss function $f(x)$ distributed across $n$ agents as given in equation. 1. Note that $f_i$ is a local loss function (for example, cross entropy loss) defined in terms of the data sampled ($d_i$) from the local dataset $D_i$ at agent $i$ with model parameters $x_i$.

$$\min_{x \in \mathbb{R}^d} f(x) = \frac{1}{n} \sum_{i=1}^{n} F_i(D_i; x_i),$$

$$and \ \ F_i(x_i, D_i) = \mathbb{E}_{d_i \in D_i}[f_i(d_i, x_i)] \ \ \forall i \tag{1}$$

This is typically achieved by combining local stochastic gradient descent Bottou (2010) with global consensus based gossip averaging Xiao & Boyd (2004). The communication topology in this peer-to-peer setup is modeled as a graph $G = ([n], E)$ with edges $\{i, j\} \in E$ if and only if agents $i$ and $j$ are connected by a communication link exchanging the messages directly. We represent $\mathcal{N}(i)$ as the neighbors of $i$ including itself. It is assumed that the graph $G$ is strongly connected with self loops i.e., there is a path from every agent to every other agent. The adjacency matrix of the graph $G$ is referred as a mixing matrix $W$ where $w_{ij}$ is the weight associated with the edge $\{i, j\}$. Note that, weight 0 indicates the absence of a direct edge between the agents. We assume that the mixing matrix is doubly-stochastic and symmetric, similar to all previous works in decentralized learning. For example, in a undirected ring topology, $w_{ij} = \frac{1}{3}$ if $j \in \{i-1, i, i+1\}$. Further, the initial models and all the hyperparameters are synchronized in the beginning of the training. Algorithm. 3 in appendix describes the flow of D-PSGD with momentum. The convergence of the Algorithm. 3 assumes the data-distribution across the agents to be Independent and Identically Distributed (IID).

## 3 NEIGHBORHOOD GRADIENT CLUSTERING

We propose the *Neighborhood Gradient Clustering (NGC)* algorithm and a compressed version of *NGC* which improve the performance of decentralized learning over non-IID data distribution. *NGC* utilizes the concepts of self-gradient and cross-gradient Esfandiari et al. (2021). The following are the definitions of self-gradient and cross-gradient.
**Self-Gradient:** For an agent $i$ with the local dataset $D_i$ and model parameters $x_i$, the self-gradient is the gradient of the loss function $f_i$ with respect to the model parameters $x_i$, evaluated on mini-batch $d_i$ sampled from dataset $D_i$.

$$g_{ii}^t = \nabla_x f_i(d_i^t; x_i^t) \tag{2}$$

**Cross-Gradient:** For an agent $i$ with model parameters $x_i$ connected to neighbor $j$ that has local dataset $D_j$, the cross-gradient is the gradient of the loss function $f_j$ with respect to the model parameters $x_i$, evaluated on mini-batch $d_j$ sampled from dataset $D_j$.

$$g_{ij}^t = \nabla_x f_j(d_j^t; x_i^t) \tag{3}$$

Note that the cross-gradient $g_{ij}$ is computed on agent $j$ using its local data after receiving the model parameters $x_i$ from its neighbouring agent $i$ and is then communicated to agent $i$.

### 3.1 THE *NGC* ALGORITHM

The flow of the Neighborhood Gradient Clustering (*NGC*) is shown in Algorithm. 1 and the form of the algorithm is similar to D-PSGD Lian et al. (2017) presented in Algorithm. 3.

---

**Algorithm 1** Neighborhood Gradient Clustering (*NGC*)

---

**Input:** Each agent $i \in [1, n]$ initializes model weights $x_i^{(0)}$, learning rate $\eta$, averaging rate $\gamma$, mixing matrix $W = [w_{ij}]_{i,j \in [1,n]}$, *NGC* mixing weight $\alpha$, and $I_{ij}$ are elements of $n \times n$ identity matrix, $\mathcal{N}(i)$ represents neighbors of $i$ including itself.

Each agent simultaneously implements the TRAIN( ) procedure
1. **procedure** TRAIN( )
2.    **for** t=0, 1, ..., $T - 1$ **do**
3.       $d_i^t \sim D_i$          // sample data from training dataset
4.       $g_{ii}^t = \nabla_x f_i(d_i^t; x_i^t)$    // compute the local self-gradients
5.       SENDRECEIVE($x_i^t$)    // shares model parameters with neighbors $\mathcal{N}(i)$
6.       **for** each neighbor $j \in \{\mathcal{N}(i) - i\}$ **do**
7.          $g_{ji}^t = \nabla_x f_i(d_i^t; x_j^t)$    // compute neighbors' cross-gradients
8.          **if** $\alpha \neq 0$ **do**
9.             SENDRECEIVE($g_{ji}^t$)    // share cross-gradients between i and j
10.         **end**
11.       **end**
12.       $\widetilde{g}_i^t = (1 - \alpha) * \sum_{j \in \mathcal{N}(i)} w_{ij} * g_{ji}^t + \alpha * \sum_{j \in \mathcal{N}(i)} w_{ij} * g_{ij}^t$    // modify local gradients
13.       $v_i^t = \beta v_i^{(t-1)} - \eta \widetilde{g}_i^t$    // momentum step
14.       $\widetilde{x}_i^t = x_i^t + v_i^t$    // update the model
15.       $x_i^{(t+1)} = \widetilde{x}_i^t + \gamma \sum_{j \in \mathcal{N}(i)} (w_{ij} - I_{ij}) * x_j^t$    // gossip averaging step
16.    **end**
17. **return**

---

The main contribution of the proposed *NGC* algorithm is the local gradient manipulation step (line 12 in Algorithm. 1). In the $t^{th}$ iteration of *NGC*, each agent $i$ calculates its self-gradient $g_{ii}$. Then, agent $i$'s model parameters are transmitted to all other agents ($j$) in its neighborhood, and the respective cross-gradients are calculated by the neighbors and transmitted back to agent $i$. At every iteration after the communication rounds, each agent $i$ has access to self-gradients ($g_{ii}$) and two sets of cross-gradients: 1) *Model-variant cross-gradients*: The derivatives that are computed locally using its local data on the neighbors' model parameters ($g_{ji}$). 2) *Data-variant cross-gradients*: The derivatives (received through communication) of its model parameters on the neighbors' dataset

$(g_{ij})$. Note that each agent $i$ computes and transmits cross-gradients $(g_{ji})$ that act as model-variant cross-gradients for $i$ and as data-variant cross-gradients for $j$. We then cluster the gradients into two groups namely: a) *Model-variant cluster* $\{g_{ji} \forall j \in \mathcal{N}(i)\}$ that includes self-gradients and model-variant cross-gradients, and b) *Data-variant cluster* $\{g_{ij} \forall j \in \mathcal{N}(i)\}$ that includes self-gradients and data-variant cross-gradients. The local gradients at each agent is replaced with the weighted average of the above defined cluster means as shown in Equation. 4, which assumes uniform mixing matrix $(w_{ij} = 1/m; m = |\mathcal{N}(i)|)$. The mean of model-variant cluster is weighted by $(1 - \alpha)$ and the mean of data-variant cluster is weighed by $\alpha$ where $\alpha \in [0, 1]$ is a hyper-parameter referred as *NGC* mixing weight.

$$\widetilde{g}_i^t = (1 - \alpha) * \underbrace{\left[\frac{1}{m} \sum_{j \in \mathcal{N}(i)} g_{ji}^t\right]}_{\text{(a) Model-variant cluster mean}} + \alpha * \underbrace{\left[\frac{1}{m} \sum_{j \in \mathcal{N}(i)} g_{ij}^t\right]}_{\text{(b) Data-variant cluster mean}} \tag{4}$$

The motivation for this modification is to reduce the variation of the computed local gradients across the agents. In IID settings, the local gradients should statistically resemble the cross-gradients and hence simple gossip averaging is sufficient to reach convergence. However, in the non-IID case, the local gradients across the agents are significantly different due to the variation in datasets and hence the model parameters on which the gradients are computed. The proposed algorithm reduces this variation in the local-gradients as it is equivalent to adding two bias terms $\epsilon$ and $\omega$ with weights $(1 - \alpha)$ and $\alpha$ respectively as shown in Equation. 5.

$$\widetilde{g}_i^t = (1 - \alpha) * \left[g_{ii}^t + \frac{1}{m} \sum_{j \in \mathcal{N}(i)} (g_{ji}^t - g_{ii}^t)\right] + \alpha * \left[g_{ii}^t + \frac{1}{m} \sum_{j \in \mathcal{N}(i)} \frac{1}{m}(g_{ij}^t - g_{ii}^t)\right]$$

$$= g_{ii}^t + (1 - \alpha) * \underbrace{\left[\frac{1}{m} \sum_{j \in \mathcal{N}(i)} (g_{ji}^t - g_{ii}^t)\right]}_{\text{model variance bias } \epsilon_i^t} + \alpha * \underbrace{\left[\frac{1}{m} \sum_{j \in \mathcal{N}(i)} \frac{1}{m}(g_{ij}^t - g_{ii}^t)\right]}_{\text{data variance bias } \omega_i^t} \tag{5}$$

$$\epsilon_i^t = \frac{1}{m} * \sum_{j \in \mathcal{N}(i)} \left(\nabla_x f(d_i^t; x_j^t) - \nabla_x f(d_i^t; x_i^t)\right)$$

$$\omega_i^t = \frac{1}{m} * \sum_{j \in \mathcal{N}(i)} \left(\nabla_x f(d_j^t; x_i^t) - \nabla_x f(d_i^t; x_i^t)\right) \quad \text{(note that } f = f_i = f_j\text{)}$$

The bias term $\epsilon$ compensates for the difference in a neighborhood's self-gradients caused due to variation in the model parameters across the neighbors. Whereas, the bias term $\omega$ compensates for the difference in a neighborhood's self-gradients caused due to variation in the data distribution across the neighbors. We hypothesis and show through our experiments that addition of these bias terms to the local gradients improves the performance of decentralized learning over non-IID data by accelerating the global convergence. Note that if we set $\alpha = 0$ in the *NGC* algorithm then it does not require an additional communication round (no communication overhead compared to D-PSGD).

## 3.2 THE COMPRESSED *NGC* ALGORITHM

The *NGC* algorithm at every iteration involves two-steps of communication with the neighbors: 1) communicate the model parameters, and 2) communicate the cross-gradients. This communication overhead can be a bottleneck in a resource-constrained environment. Hence we propose a compressed version of *NGC* using Error Feedback SGD (EF-SGD) Karimireddy et al. (2019) to compress gradients. We compress the error-compensated self-gradients and cross-gradients from 32 bits (floating point precision of arithmetic operations) to 1 bit by using scaled signed gradients. The error between the compressed and non-compressed gradient of the current iteration ($e_{ji}^t$ in the algorithm) is added as feedback to the gradients in the next iteration before compression. The pseudo code for *CompNGC* is shown in Algorithm. 2.

---

**Algorithm 2** Compressed Neighborhood Gradient Clustering (*CompNGC*)

---

**Input:** Each agent $i \in [1, n]$ initializes model weights $x_i^{(0)}$, learning rate $\eta$, averaging rate $\gamma$, dimension of the gradient $d$, mixing matrix $W = [w_{ij}]_{i,j \in [1,n]}$, *NGC* mixing weight $\alpha$, and $I_{ij}$ are elements of $n \times n$ identity matrix.

Each agent simultaneously implements the TRAIN( ) procedure
1. **procedure** TRAIN( )
2.     **for** t=$0, 1, \ldots, T - 1$ **do**
3.         $d_i^t \sim D_i$                   // sample data from training dataset
4.         $g_{ii}^t = \nabla_x f_i(d_i^t; x_i^t)$         // compute the local self-gradients
5.         $p_{ii}^t = g_{ii}^t + e_{ii}^t$         // error compensation for self-gradients
6.         $\delta_{ii}^t = (\|p_{ii}^t\|_1/d) sgn(p_{ii})$     // compress the compensated self-gradients
7.         $e_{ii}^{t+1} = p_{ii}^t - \delta_{ii}^t$         // update the error variable
8.         SENDRECEIVE($x_i^t$)         // share model parameters with neighbors $N(i)$
9.         **for** each neighbor $j \in \{N(i) - i\}$ **do**
10.            $g_{ji}^t = \nabla_x f_i(d_i^t; x_j^t)$      // compute neighbors' cross-gradients
11.            $p_{ji}^t = g_{ji}^t + e_{ji}^t$        // error compensation for cross-gradients
12.            $\delta_{ji}^t = (\|p_{ji}^t\|_1/d) sgn(p_{ji})$    // compress the compensated cross-gradients
13.            $e_{ji}^{t+1} = p_{ji}^t - \delta_{ji}^t$      // update the error variable
14.            **if** $\alpha \neq 0$ **do**
15.                SENDRECEIVE($\delta_{ji}^t$)     // share compressed cross-gradients between $i$ and $j$
16.            **end**
17.         **end**
18.         $\widetilde{g}_i^t = (1 - \alpha) * \sum_{j \in \mathcal{N}(i)} w_{ij} * \delta_{ji}^t + \alpha * \sum_{j \in \mathcal{N}(i)} w_{ij} * \delta_{ij}^t$   // modify local gradients
19.         $v_i^t = \beta v_i^{(t-1)} - \eta \widetilde{g}_i^t$       // momentum step
20.         $\widetilde{x}_i^t = x_i^t + v_i^t$         // update the model
21.         $x_i^{(t+1)} = \widetilde{x}_i^t + \gamma \sum_{j \in \mathcal{N}(i)} (w_{ij} - I_{ij}) * x_j^t$   // gossip averaging step
22.     **end**
23. **return**

---

## 4   EXPERIMENTS

In this section, we analyze the performance of the proposed *NGC* and *CompNGC* techniques and compare them with the baseline D-PSGD algorithm Lian et al. (2017) and state-of-the-art CGA and CompCGA methods Esfandiari et al. (2021).

***Experimental Setup:*** The efficiency of the proposed method is demonstrated through our experiments on diverse set of datasets, model architectures, tasks, topologies and numbers of agents. We present the analysis on – (a) Datasets (Appendix A.2): vision datasets (CIFAR-10, CIFAR-100, Fashion MNIST and Imagenette Husain (2018)) and language datasets (AGNews Zhang et al. (2015)). (b) Model architectures (Appendix A.3): 5-layer CNN, VGG-11, ResNet-20, LeNet-5, MobileNet-V2, ResNet-18, BERT$_{mini}$ and DistilBERT$_{base}$ (c) Tasks: Image and Text Classification. (d) Topologies: Ring, Chain and Torus. (e) Number of agents: varying from 4 to 20. Note that we use low resolution ($32 \times 32$) images of Imagenette dataset for the experiments in Table. 2. The results for high resolution ($224 \times 224$) Imagenette are presented in Table. 3. We consider an extreme case of non-IID distribution where no two neighboring agents have same class. This is referred as complete label-wise skew or 100% label-wise non-IIDness Hsieh et al. (2020). In particular for a 10-class dataset such as CIFAR-10 - each agent in a 5 agents system has data from 2 distinct classes, each agent in a 10 agents system has data from an unique class. For a 20 agent system two agents that are maximally apart share the samples belonging to a class. We report the test accuracy of the consensus model averaged over three randomly chosen seeds. The details of the hyperparameters for all the experiments are present in Appendix. A.4. We compare the propose method with iso-communication baselines. The experiments on *NGC* ($\alpha = 0$) are compared with D-PSGD, *NGC* with CGA, and *CompNGC* with CompCGA. The communication cost for each experiment in this section is presented in Appendix A.6.

**Results:** We evaluate variants of *NGC* and *CompNGC* and compare them with respective baselines in Table. 1, for training different models trained on CIFAR-10 over various graph sizes and topologies. We observe that *NGC* with $\alpha = 0$ consistently outperforms D-PSGD for all models, graph sizes and topologies with a significant performance gain varying from $3-35\%$. Our experiments show the superiority of *NGC* and *CompNGC* over CGA and CompCGA respectively. The performance gains are more pronounced when considering larger graphs (with 20 agents) and compact models such as ResNet-20. We further demonstrate the generalizability of the proposed method by evaluating it on various image datasets such as Fashion MNIST, Imagenette and on challenging dataset such as CIFAR-100. Table. 2, 3 show that *NGC* with $\alpha = 0$ outperforms D-PSGD by $2-13\%$ across various datasets where as *NGC* and *CompNGC* remain competitive with an average improvement of $\sim 1\%$. To show the effectiveness of the proposed method across different modalities, we present results on text classification task in Table 4. We train on BERT$_{mini}$ model with AGNews dataset distributed over 4 and 8 agents and a larger transformer model (DistilBert$_{base}$) distributed over 4 agents. For NGC $\alpha = 0$ we see an maximum improvement of 2.1% over the baseline D-PSGD algorithm. Even for the text classification task, we observe *NGC* and *CompNGC* to be competitive with CGA and CompCGA methods. These observations are consistent with the results on the image classification tasks. Finally, through these exhaustive set of experiments, we demonstrate that the weighted averaging of data-variant and model-variant cross-gradients can be served as a simple plugin to boost the performance of decentralized learning over label-wise non-IID data. Further, locally available model-variant cross-gradients information at each agent can be efficiently utilized to improve the decentralized learning with no communication overhead.

Table 1: Average test accuracy comparisons for CIFAR-10 with non-IID data using various architectures and graph topologies. The results are averaged over three seeds where std is indicated.

| Method | Agents | 5layer CNN Ring | 5layer CNN Torus | VGG-11 Ring | ResNet-20 Ring |
|---|---|---|---|---|---|
| D-PSGD | 5 | $76.00 \pm 1.44$ | - | $67.04 \pm 5.36$ | $82.13 \pm 0.84$ |
| | 10 | $47.68 \pm 3.20$ | $55.34 \pm 6.32$ | $44.14 \pm 3.30$ | $31.66 \pm 6.01$ |
| | 20 | $44.85 \pm 1.94$ | $50.12 \pm 1.91$ | $38.92 \pm 2.99$ | $31.94 \pm 2.91$ |
| *NGC* (ours) ($\alpha = 0$) | 5 | $\mathbf{82.20 \pm 0.34}$ | - | $\mathbf{79.27 \pm 0.39}$ | $\mathbf{85.88 \pm 0.58}$ |
| | 10 | $\mathbf{67.43 \pm 1.15}$ | $\mathbf{73.84 \pm 0.33}$ | $\mathbf{59.92 \pm 2.12}$ | $\mathbf{66.02 \pm 2.86}$ |
| | 20 | $\mathbf{58.80 \pm 1.30}$ | $\mathbf{64.55 \pm 1.16}$ | $\mathbf{52.70 \pm 1.63}$ | $\mathbf{50.74 \pm 2.36}$ |
| CGA | 5 | $82.20 \pm 0.43$ | - | $84.41 \pm 0.22$ | $87.52 \pm 0.50$ |
| | 10 | $72.96 \pm 0.40$ | $76.04 \pm 0.62$ | $\mathbf{79.66 \pm 0.46}$ | $79.98 \pm 1.23$ |
| | 20 | $69.88 \pm 0.84$ | $73.21 \pm 0.27$ | $79.30 \pm 0.12$ | $75.13 \pm 1.56$ |
| *NGC* (ours) | 5 | $\mathbf{83.36 \pm 0.65}$ | - | $\mathbf{85.15 \pm 0.58}$ | $\mathbf{88.52 \pm 0.19}$ |
| | 10 | $\mathbf{75.34 \pm 0.30}$ | $\mathbf{78.53 \pm 0.56}$ | $79.55 \pm 0.30$ | $\mathbf{84.02 \pm 0.44}$ |
| | 20 | $\mathbf{73.36 \pm 0.88}$ | $\mathbf{75.11 \pm 0.07}$ | $\mathbf{79.43 \pm 0.62}$ | $\mathbf{81.26 \pm 0.69}$ |
| CompCGA | 5 | $82.00 \pm 0.25$ | - | $83.65 \pm 0.41$ | $86.73 \pm 0.34$ |
| | 10 | $71.41 \pm 0.94$ | $75.95 \pm 0.41$ | $73.96 \pm 0.31$ | $73.63 \pm 0.55$ |
| | 20 | $68.15 \pm 0.79$ | $71.71 \pm 0.54$ | $73.72 \pm 2.74$ | $66.34 \pm 0.98$ |
| *CompNGC* (ours) | 5 | $\mathbf{82.91 \pm 0.21}$ | - | $\mathbf{84.03 \pm 0.32}$ | $\mathbf{87.56 \pm 0.34}$ |
| | 10 | $\mathbf{74.36 \pm 0.42}$ | $\mathbf{77.82 \pm 0.20}$ | $\mathbf{77.02 \pm 0.14}$ | $\mathbf{78.50 \pm 0.98}$ |
| | 20 | $\mathbf{71.46 \pm 0.85}$ | $\mathbf{73.62 \pm 0.74}$ | $\mathbf{73.76 \pm 0.20}$ | $\mathbf{72.62 \pm 0.71}$ |

**Analysis:** We show the convergence characteristics of the proposed algorithm over IID and Non-IID data distributions in Figure. 1a, and 1b respectively. For Non-IID distribution, we observe that there is a slight difference in convergence (as expected) with slower rate for sparser topology (ring graph) compared to denser counterpart (fully connected graph). Figure. 1c shows the comparison of the convergence characteristics of the *NGC* algorithm with the current state-of-the-art CGA algorithm. We observe that *NGC* has lower validation loss than CGA for same decentralized setup. Analysis for 10 agents is presented in Appendix A.5. The change in average validation accuracy with *NGC* mixing weight $\alpha$ is shown in Figure. 2. We observe that $\alpha$ close to 1 has the best performance as the model variance bias is taken care in the gossip averaging step. We also plot the model variance and data variance bias terms for both *NGC* and CGA techniques as shown in Figure. 3a, and 3b respectively. We observe that both the model variance and the data variance bias for *NGC* are

Table 2: Average test accuracy comparisons for various datasets with non-IID sampling trained over undirected ring topology. The results are averaged over three seeds where std is indicated.

| Method | Agents | Fashion MNIST (LeNet-5) | CIFAR-100 (ResNet-20) | Imagenette (MobileNet-V2) |
|---|---|---|---|---|
| D-PSGD | 5 | $86.43 \pm 0.14$ | $44.66 \pm 5.23$ | $47.09 \pm 9.20$ |
|  | 10 | $75.49 \pm 0.32$ | $19.03 \pm 13.27$ | $32.81 \pm 2.18$ |
| *NGC (ours)* | 5 | $\mathbf{88.49 \pm 0.18}$ | $\mathbf{55.96 \pm 0.95}$ | $\mathbf{60.15 \pm 2.17}$ |
| ($\alpha = 0$) | 10 | $\mathbf{82.85 \pm 0.24}$ | $\mathbf{35.34 \pm 0.32}$ | $\mathbf{36.13 \pm 1.97}$ |
| CGA | 5 | $90.03 \pm 0.39$ | $56.43 \pm 2.39$ | $72.82 \pm 1.25$ |
|  | 10 | $\mathbf{87.61 \pm 0.30}$ | $53.61 \pm 1.07$ | $61.97 \pm 0.58$ |
| *NGC (ours)* | 5 | $\mathbf{90.61 \pm 0.18}$ | $\mathbf{56.50 \pm 3.23}$ | $\mathbf{74.49 \pm 0.93}$ |
|  | 10 | $87.24 \pm 0.23$ | $\mathbf{53.77 \pm 0.15}$ | $\mathbf{64.06 \pm 1.11}$ |
| CompCGA | 5 | $90.45 \pm 0.34$ | $55.74 \pm 0.33$ | $72.76 \pm 0.44$ |
|  | 10 | $81.62 \pm 0.37$ | $38.84 \pm 0.54$ | $59.92 \pm 0.72$ |
| *CompNGC (ours)* | 5 | $\mathbf{90.48 \pm 0.19}$ | $\mathbf{57.51 \pm 0.48}$ | $\mathbf{72.91 \pm 1.06}$ |
|  | 10 | $\mathbf{83.38 \pm 0.39}$ | $\mathbf{43.07 \pm 0.32}$ | $\mathbf{61.91 \pm 2.10}$ |

Table 3: Average test accuracy comparisons for full resolution ($224 \times 224$) Imagenette dataset with non-IID data trained on ResNet-18 over 5 agents. The results are averaged over three seeds.

| Method | Ring topology | Chain topology |
|---|---|---|
| D-PSGD | $65.43 \pm 4.60$ | $42.02 \pm 1.25$ |
| *NGC $\alpha = 0$ (ours)* | $\mathbf{73.15 \pm 0.38}$ | $\mathbf{47.87 \pm 0.99}$ |
| CGA | $85.00 \pm 0.67$ | $65.96 \pm 1.84$ |
| *NGC (ours)* | $\mathbf{85.85 \pm 0.60}$ | $\mathbf{67.77 \pm 1.76}$ |
| CompCGA | $84.65 \pm 0.57$ | $\mathbf{62.93 \pm 1.33}$ |
| *CompNGC (ours)* | $\mathbf{85.44 \pm 0.10}$ | $62.64 \pm 0.85$ |

significantly lower than CGA. This is because CGA gives more importance to self-gradients as it updates in the descent direction that is close to self-gradients and is positively correlated to data-variant cross-gradients. In contrast, *NGC* accounts for the biases directly and gives equal importance to self-gradients and data-variant cross-gradients, thereby achieving superior performance.

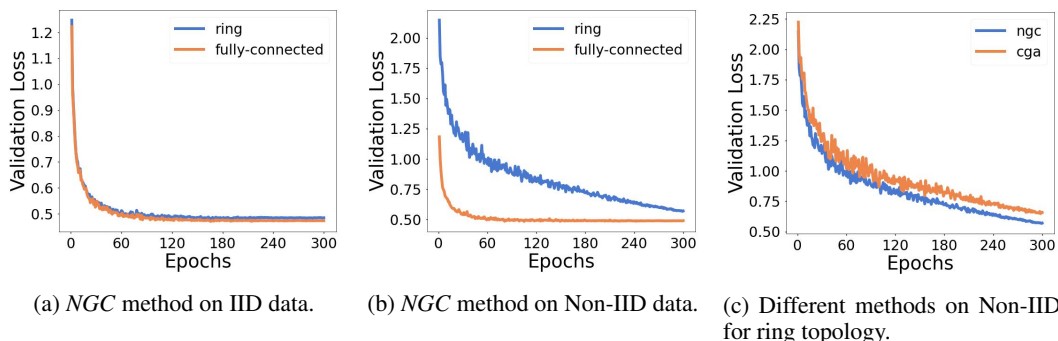

(a) *NGC* method on IID data.  (b) *NGC* method on Non-IID data.  (c) Different methods on Non-IID for ring topology.

Figure 1: Average Validation loss during training of 5 agents on CIFAR-10 with a 5 layer CNN.

***Hardware Benefits:*** The proposed *NGC* algorithm is superior in terms of memory and compute efficiency (see Table. 5), while having equal communication cost as compared to *CGA*. Since *NGC* involves weighted averaging, we do not need any additional buffer to store the cross-gradients. Weighted cross-gradients can be added to the the self-gradient buffer. CGA stores all the cross-gradients in a matrix form for quadratic programming projection of local gradient. Therefore, *NGC* has no memory overhead compared to the baseline D-PSGD algorithm, while CGA requires additional memory equivalent to the number of neighbors times model size. Moreover, the quadratic

Table 4: Average test accuracy comparisons for AGNews dataset with non-IID data trained over undirected ring topology. The results are averaged over three seeds where std is indicated.

| Method | BERT$_{mini}$ | | DistilBERT$_{base}$ |
|---|---|---|---|
| | Agents = 4 | Agents = 8 | Agents = 4 |
| D-PSGD | $89.21 \pm 0.41$ | $85.48 \pm 0.71$ | $91.54 \pm 0.07$ |
| NGC $\alpha = 0$ (ours) | $\mathbf{89.40 \pm 0.13}$ | $\mathbf{87.58 \pm 0.07}$ | $\mathbf{91.70 \pm 0.11}$ |
| CGA | $91.43 \pm 0.11$ | $\mathbf{89.15 \pm 0.45}$ | $93.42 \pm 0.04$ |
| NGC (ours) | $\mathbf{92.24 \pm 0.29}$ | $89.02 \pm 0.39$ | $\mathbf{94.11 \pm 0.01}$ |
| CompCGA | $91.05 \pm 0.29$ | $88.91 \pm 0.25$ | $\mathbf{93.54 \pm 0.03}$ |
| CompNGC (ours) | $\mathbf{91.24 \pm 0.43}$ | $\mathbf{89.01 \pm 0.13}$ | $93.50 \pm 0.16$ |

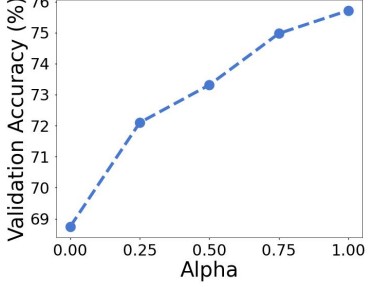
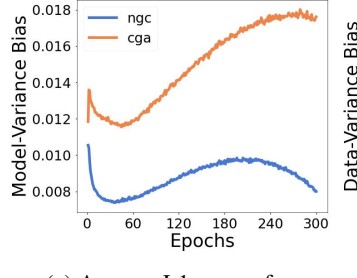
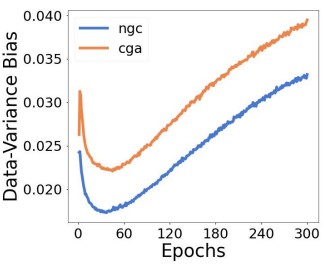

(a) Average L1 norm of $\epsilon$.  (b) Average L1 norm of $\omega$.

Figure 2: *NGC* mixing weight $\alpha$ variation for 10 agents trained on ring graph with 5 layer CNN.

Figure 3: Average L1 norm of model variance bias and data variance bias for 5 agents trained on ring graph with 5 layer CNN.

programming projection step Goldfarb & Idnani (1983) in CGA is much more expensive in terms of compute and latency as compared to weighted averaging step of cross-gradients in *NGC*. Our experiments clearly show *NGC* is superior to CGA in terms of test accuracy, memory efficiency, compute efficiency and latency.

Table 5: Comparison of communication, memory and compute overheads per mini-batch compared to D-PSGD. $m_s$: model size, $N_i$: number of neighbors, $b$: floating point precision, QP: compute for Quadratic Programming, FP: compute for Forward Pass. (see Appendix A.7 for more details)

| Method | Comm. | Memory | Compute |
|---|---|---|---|
| CGA | $\mathcal{O}(m_s N_i)$ | $\mathcal{O}(N_i m_s)$ | $\mathcal{O}(3N_i FP + QP)$ |
| CompCGA | $\mathcal{O}(\frac{m_s N_i}{b})$ | $\mathcal{O}((1 + \frac{1}{b})N_i m_s)$ | $\mathcal{O}(3N_i FP + QP)$ |
| NGC $\alpha = 0$ (ours) | $0$ | $0$ | $\mathcal{O}(3N_i FP + m_s N_i)$ |
| NGC (ours) | $\mathcal{O}(m_s N_i)$ | $0$ | $\mathcal{O}(3N_i FP + m_s N_i)$ |
| CompNGC (ours) | $\mathcal{O}(\frac{m_s N_i}{b})$ | $\mathcal{O}(N_i m_s)$ | $\mathcal{O}(3N_i FP + m_s N_i)$ |

## 5  CONCLUSION

Enabling decentralized training over non-IID data is key for ML applications to efficiently leverage the humongous amounts of user-generated private data. In this paper, we propose the *Neighborhood Gradient Clustering* (*NGC*) algorithm that improves decentralized learning over non-IID data distributions. Further, we present a compressed version of our algorithm (*CompNGC*) to reduce the communication overhead associated with *NGC*. We validate the performance of the proposed techniques (*NGC* and *CompNGC*) over different model architectures, graph sizes and topologies. Finally, we compare the proposed algorithms with the current state-of-the-art decentralized learning algorithm over non-IID data and show superior performance of our algorithm with significantly less compute and memory requirements.

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

# A  APPENDIX

## A.1  DECENTRALIZED LEARNING SETUP

The traditional decentralized learning algorithm (d-psgd) is described as Algorithm. 3. For the decentralized setup, we use undirected ring and undirected torus graph topologies with uniform mixing matrix. The undirected ring topology for any graph-size has 3 peers per agent including itself and each edge has a weight of $\frac{1}{3}$. The undirected torus topology with 10 agents has 4 peers per agent including itself and each edge has a weight of $\frac{1}{4}$. The undirected torus topology 20 agents has 5 peers per agent including itself and each edge has a weight of $\frac{1}{5}$.

---

**Algorithm 3** Decentralized Peer-to-Peer Training (*D-PSGD* with momentum)

---

**Input:** Each agent $i \in [1, n]$ initializes model weights $x_i^{(0)}$, learning rate $\eta$, averaging rate $\gamma$, mixing matrix $W = [w_{ij}]_{i,j \in [1,n]}$, and $I_{ij}$ are elements of $n \times n$ identity matrix.

Each agent simultaneously implements the TRAIN( ) procedure
1. **procedure** TRAIN( )
2.     **for** t=$0, 1, \ldots, T - 1$ **do**
3.         $d_i^t \sim D_i$                                  // sample data from training dataset.
4.         $g_i^t = \nabla_x f_i(d_i^t; x_i^t)$                          // compute the local gradients
5.         $v_i^t = \beta v_i^{(t-1)} - \eta g_i^t$                        // momentum step
6.         $\widetilde{x}_i^t = x_i^t + v_i^t$                          // update the model
7.         SENDRECEIVE($\widetilde{x}_i^t$)                    // share model parameters with neighbors $N(i)$.
8.         $x_i^{(t+1)} = \widetilde{x}_i^t + \gamma \sum_{j \in \mathcal{N}(i)} (w_{ij} - I_{ij}) * \widetilde{x}_j^t$        // gossip averaging step
9.     **end**
10. **return**

---

We highlight the various assumptions required for the convergence of the proposed *NGC* and *Comp-NGC* algorithm.

1. **Graph Structure:** The graph topology of the agents is strongly connected and symmetric.
2. **Doubly stochastic mixing matrix:** The mixing matrix $W$ is a real doubly stochastic matrix i.e. $\mathbf{1}^T W = W \mathbf{1} = W$, where $\mathbf{1}$ is the vector of all $1's$.
3. **Lipschitz Gradients:** Local loss functions $f_i(.)$ has L-lipschitz gradients for all $i \in [1, n]$ i.e. $||F_i(x) - F_i(y)|| \leq L||x - y|| \quad \forall x, y \in \mathbb{R}^d$
4. **Bounded Variance:** The variance of the stochastic gradients are assumed to be bounded. $\mathbb{E}_{d_i \sim D_i} ||\nabla f_i(x; d_i) - \nabla F_i(x)||^2 \leq \sigma^2$, (Inner variance). $\frac{1}{n} \sum_{i=1}^{n} ||\nabla F_i(x) - \nabla f(x)||^2 \leq \zeta^2 \ \forall i, x$ (Outer variance).
5. **Initialization:** The model parameters on all the agents are initialized to the same random values.

It may be noted that assumptions 1-5 are similar to that of CGA algorithm Esfandiari et al. (2021) and are commonly used in most decentralized training algorithms.

## A.2  DATASETS

In this section, we give a brief description of the datasets used in our experiments. We use a diverse set of datasets each originating from a different distribution of images to show the generalizability of the proposed techniques.

**CIFAR-10:** CIFAR-10 Krizhevsky et al. (2014) is a image classification dataset with 10 classes. The image samples are colored (3 input channels) and have a resolution of $32 \times 32$. There are $50,000$ training samples with 5000 samples per class and $10,000$ test samples with 1000 samples per class.

**CIFAR-100:** CIFAR-100 Krizhevsky et al. (2014) is a image classification dataset with 100 classes. The image samples are colored (3 input channels) and have a resolution of $32 \times 32$. There are $50,000$

training samples with 500 samples per class and $10,000$ test samples with 100 samples per class. CIFAR-100 classification is a harder task compared to CIFAR-10 as it has 100 classes with very less samples per class to learn from.

**Fashion MNIST:** Fashion MNIST Xiao et al. (2017) is a image classification dataset with 10 classes. The image samples are in grey scale (1 input channel) and have a resolution of $28 \times 28$. There are $60,000$ training samples with 6000 samples per class and $10,000$ test samples with 1000 samples per class.

**Imagenette:** Imagenette Husain (2018) is a 10 class subset of the ImageNet dataset. The image samples are in colored (3 input channel) and have a resolution of $224 \times 224$. There are 9469 training samples with roughly 950 samples per class and 3925 test samples. We conduct our experiments on two different resolutions of Imagenette dataset – a) a resized low resolution of $32 \times 32$ and, b) a full resolution of $224 \times 224$. The Imagenette experiments reported in Table. 2 use the low resolution images where as experiments in Table. 3 use the full resolution images.

**AGNews:** We use AGNews Zhang et al. (2015) dataset for Natural Language Processing (NLP) task. This is a text classification dataset where the given text news is classified into 4 classes, namely "World", "Sport", "Business" and "Sci/Tech". The dataset has a total of 120000 and 7600 samples for training and testing respectively, which are equally distributed across each class.

### A.3 NETWORK ARCHITECTURE

We replace ReLU+BatchNorm layers of all the model architectures with EvoNorm-S0 Liu et al. (2020) as it was shown to be better suited for decentralized learning over non-IID distributions Lin et al. (2021).

**5 layer CNN:** The 5 layer CNN consists of 4 convolutional with EvoNorm-S0 Liu et al. (2020) as activation-normalization layer, 3 max pooling layer and one linear layer. In particular, it has 2 convolutional layers with 32 filters, a max pooling layer, then 2 more convolutional layers with 64 filters each followed by another max pooling layer and a dense layer with 512 units. It has a total of $76K$ trainable parameters.

**VGG-11:** We modify the standard VGG-11 Simonyan & Zisserman (2014) architecture by reducing the number of filters in each convolutional layer by $4\times$ and use only one dense layer with 128 units. Each convolutional layer is followed by EvoNorm-S0 as activation-normalization layer and it has $0.58M$ trainable parameters.

**ResNet-20:** For ResNet-20 He et al. (2016), we use the standard architecture with $0.27M$ trainable parameters except that BatchNorm+ReLU layers are replaced by EvoNorm-S0.

**LeNet-5:** For LeNet-5 LeCun et al. (1998), we use the standard architecture with $61,706$ trainable parameters.

**MobileNet-V2:** We use the the standard MobileNet-V2 Sandler et al. (2018) architecture used for CIFAR dataset with $2.3M$ parameters except that BatchNorm+ReLU layers are replaced by EvoNorm-S0.

**ResNet-18:** For ResNet-18 He et al. (2016), we use the standard architecture with $11M$ trainable parameters except that BatchNorm+ReLU layers are replaced by EvoNorm-S0.

**BERT$_{\text{mini}}$:** For BERT$_{\text{mini}}$ Devlin et al. (2018) we use the standard model from the paper. We restrict the sequence length of the model to 128. The model used in the work hence has $11.07M$ parameters.

**DistilBERT$_{\text{base}}$:** For DistilBERT$_{\text{base}}$ Sanh et al. (2019) we use the standard model from the paper. We restrict the sequence length of the model to 128. The model used in the work hence has $66.67M$ parameters.

### A.4 HYPER-PARAMETERS

All the experiments were run for three randomly chosen seeds. We decay the learning-rate by 10x after 50% and 75% of the training, unless mentioned otherwise.

Table 6: Hyper-parameters used for CIFAR-10 with non-IID data distribution using 5-layer CNN model architecture presented in Table 1

| Method | Agents (n) | Ring $(\alpha, \beta, \eta, \gamma)$ | Torus $(\alpha, \beta, \eta, \gamma)$ |
|---|---|---|---|
| D-PSGD | 5 | $(-, 0.0, 0.1, 1.0)$ | $-$ |
| | 10 | $(-, 0.0, 0.1, 1.0)$ | $(-, 0.0, 0.1, 1.0)$ |
| | 20 | $(-, 0.0, 0.1, 1.0)$ | $(-, 0.0, 0.1, 1.0)$ |
| *NGC* (ours) | 5 | $(0.0, 0.0, 0.1, 1.0)$ | $-$ |
| $(\alpha = 0)$ | 10 | $(0.0, 0.0, 0.1, 1.0)$ | $(0.0, 0.0, 0.1, 1.0)$ |
| | 20 | $(0.0, 0.0, 0.1, 1.0)$ | $(0.0, 0.0, 0.1, 1.0)$ |
| CGA | 5 | $(-, 0.9, 0.01, 0.1)$ | $-$ |
| | 10 | $(-, 0.9, 0.01, 0.5)$ | $(-, 0.9, 0.01, 0.1)$ |
| | 20 | $(-, 0.9, 0.01, 0.5)$ | $(-, 0.9, 0.01, 0.1)$ |
| *NGC* (ours) | 5 | $(1.0, 0.9, 0.01, 0.1)$ | $-$ |
| | 10 | $(1.0, 0.9, 0.01, 0.5)$ | $(1.0, 0.9, 0.01, 0.1)$ |
| | 20 | $(1.0, 0.9, 0.01, 0.5)$ | $(1.0, 0.9, 0.01, 0.1)$ |
| CompCGA | 5 | $(-, 0.9, 0.01, 0.1)$ | $-$ |
| | 10 | $(-, 0.9, 0.01, 0.5)$ | $(-, 0.9, 0.01, 0.1)$ |
| | 20 | $(-, 0.9, 0.01, 0.5)$ | $(-, 0.9, 0.01, 0.1)$ |
| *CompNGC* (ours) | 5 | $(1.0, 0.9, 0.01, 0.1)$ | $-$ |
| | 10 | $(1.0, 0.9, 0.01, 0.5)$ | $(1.0, 0.9, 0.01, 0.1)$ |
| | 20 | $(1.0, 0.9, 0.01, 0.5)$ | $(1.0, 0.9, 0.01, 0.1)$ |

Table 7: Hyper-parameters used for CIFAR-10 with non-IID data distribution using ResNet and VGG-11 model architecture presented in Table 1

| Method | Agents (n) | VGG-11 $(\alpha, \beta, \eta, \gamma)$ | ResNet $(\alpha, \beta, \eta, \gamma)$ |
|---|---|---|---|
| D-PSGD | 5 | $(-, 0.0, 0.01, 1.0)$ | $(-, 0.0, 0.1, 1.0)$ |
| | 10 | $(-, 0.0, 0.01, 1.0)$ | $(-, 0.0, 0.1, 1.0)$ |
| | 20 | $(-, 0.0, 0.01, 1.0)$ | $(-, 0.0, 0.1, 1.0)$ |
| *NGC* (ours) | 5 | $(0.0, 0.0, 0.01, 1.0)$ | $(0.0, 0.0, 0.1, 1.0)$ |
| $(\alpha = 0)$ | 10 | $(0.0, 0.0, 0.01, 1.0)$ | $(0.0, 0.0, 0.1, 1.0)$ |
| | 20 | $(0.0, 0.0, 0.01, 1.0)$ | $(0.0, 0.0, 0.1, 1.0)$ |
| CGA | 5 | $(-, 0.9, 0.1, 0.5)$ | $(-, 0.9, 0.1, 1.0)$ |
| | 10 | $(-, 0.9, 0.1, 0.5)$ | $(-, 0.9, 0.1, 1.0)$ |
| | 20 | $(-, 0.9, 0.1, 0.5)$ | $(-, 0.9, 0.1, 1.0)$ |
| *NGC* (ours) | 5 | $(1.0, 0.9, 0.1, 0.5)$ | $(1.0, 0.9, 0.1, 1.0)$ |
| | 10 | $(1.0, 0.9, 0.1, 0.5)$ | $(1.0, 0.9, 0.1, 1.0)$ |
| | 20 | $(1.0, 0.9, 0.1, 0.5)$ | $(1.0, 0.9, 0.1, 1.0)$ |
| CompCGA | 5 | $(-, 0.9, 0.01, 0.1)$ | $(-, 0.9, 0.01, 0.1)$ |
| | 10 | $(-, 0.9, 0.01, 0.1)$ | $(-, 0.9, 0.01, 0.1)$ |
| | 20 | $(-, 0.9, 0.01, 0.1)$ | $(-, 0.9, 0.01, 0.1)$ |
| *CompNGC* (ours) | 5 | $(1.0, 0.9, 0.01, 0.1)$ | $(1.0, 0.9, 0.01, 0.1)$ |
| | 10 | $(1.0, 0.9, 0.01, 0.1)$ | $(1.0, 0.9, 0.01, 0.1)$ |
| | 20 | $(1.0, 0.9, 0.01, 0.1)$ | $(1.0, 0.9, 0.01, 0.1)$ |

**Hyper-parameters for CIFAR-10 on 5 layer CNN:** All the experiments that involve 5layer CNN model (Table. 1) have stopping criteria set to 100 epochs. We decay the learning rate by $10\times$ in a multiple steps at $50^{th}$ and $75^{th}$ epoch. Table 6 presents the $\alpha$, $\beta$, $\eta$, and $\gamma$ corresponding to the ngc mixing weight, momentum, learning-rate and gossip averaging rate. For all the experiments, we use a mini-batch size of 32 per agent. The stopping criteria is a fixed number of epochs. We have used Nesterov momentum of 0.9 for all CGA and *NGC* experiments where as D-PSGD and *NGC* with $\alpha = 0$ has no momentum.

**Hyper-parameters for CIFAR-10 on VGG-11 and ResNet-20:** All the experiments for CIFAR-10 dataset trained on VGG-11 and ResNet-20 architectures (Table. 1) have stopping criteria set to 200 epochs. We decay the learning rate by $10\times$ in a multiple steps at $100^{th}$ and $150^{th}$ epoch. Table 7 presents the $\alpha$, $\beta$, $\eta$, and $\gamma$ corresponding to the ngc mixing weight, momentum, learning-rate and gossip averaging rate. For all the experiments, we use a mini-batch size of 32 per agent.

**Hyper-parameters used for Table. 2:** All the experiments in Table. 2 have stopping criteria set to 100 epochs. We decay the learning rate by $10\times$ in a multiple steps at $50^{th}$ and $75^{th}$ epoch. Table 8 presents the $\alpha$, $\beta$, $\eta$, and $\gamma$ corresponding to the ngc mixing weight, momentum, learning-rate and gossip averaging rate. For all the experiments related to Fashion MNIST and Imagenette (low resolution of $(32 \times 32)$), we use a mini-batch size of 32 per agent. For all the experiments related to CIFAR-100, we use a mini-batch size of 20 per agent.

Table 8: Hyper-parameters used for Table. 2

| Method | Agents (n) | Fashion MNIST $(\alpha, \beta, \eta, \gamma)$ | CIFAR-100 $(\alpha, \beta, \eta, \gamma)$ | Imagenette $(\alpha, \beta, \eta, \gamma)$ |
|---|---|---|---|---|
| D-PSGD | 5 | $(-, 0.0, 0.01, 1.0)$ | $(-, 0.0, 0.1, 1.0)$ | $(-, 0.0, 0.1, 1.0)$ |
| | 10 | $(-, 0.0, 0.01, 1.0)$ | $(-, 0.0, 0.1, 1.0)$ | $(-, 0.0, 0.1, 1.0)$ |
| *NGC* (ours) | 5 | $(0.0, 0.0, 0.01, 1.0)$ | $(0.0, 0.0, 0.1, 1.0)$ | $(-, 0.0, 0.1, 1.0)$ |
| ($\alpha = 0$) | 10 | $(0.0, 0.0, 0.01, 1.0)$ | $(0.0, 0.0, 0.1, 1.0)$ | $(-, 0.0, 0.1, 1.0)$ |
| CGA | 5 | $(-, 0.9, 0.01, 1.0)$ | $(-, 0.9, 0.1, 1.0)$ | $(0.0, 0.0, 0.01, 0.5)$ |
| | 10 | $(-, 0.9, 0.01, 1.0)$ | $(-, 0.9, 0.1, 0.5)$ | $(0.0, 0.0, 0.01, 0.5)$ |
| *NGC* (ours) | 5 | $(1.0, 0.9, 0.01, 1.0)$ | $(1.0, 0.9, 0.1, 1.0)$ | $(0.0, 0.0, 0.01, 0.5)$ |
| | 10 | $(1.0, 0.9, 0.01, 1.0)$ | $(1.0, 0.9, 0.1, 0.5)$ | $(0.0, 0.0, 0.01, 0.5)$ |
| CompCGA | 5 | $(-, 0.9, 0.01, 0.1)$ | $(-, 0.9, 0.01, 0.1)$ | $(0.0, 0.0, 0.01, 0.1)$ |
| | 10 | $(-, 0.9, 0.01, 0.1)$ | $(-, 0.9, 0.01, 0.1)$ | $(0.0, 0.0, 0.01, 0.5)$ |
| *CompNGC* (ours) | 5 | $(1.0, 0.9, 0.01, 0.1)$ | $(1.0, 0.9, 0.01, 0.1)$ | $(0.0, 0.0, 0.01, 0.1)$ |
| | 10 | $(1.0, 0.9, 0.01, 0.1)$ | $(1.0, 0.9, 0.01, 0.1)$ | $(0.0, 0.0, 0.01, 0.5)$ |

**Hyper-parameters used for Table. 3:** All the experiments in Table. 3 have stopping criteria set to 100 epochs. We decay the learning rate by $10\times$ at $50^{th}, 75^{th}$ epoch. Table 9 presents the $\alpha$, $\beta$, $\eta$, and $\gamma$ corresponding to the ngc mixing weight, momentum, learning-rate and gossip averaging rate. For all the experiments, we use a mini-batch size of 32 per agent.

Table 9: Hyper-parameters used for Table. 3

| Method | Ring topology $(\alpha, \beta, \eta, \gamma)$ | Chain topology $(\alpha, \beta, \eta, \gamma)$ |
|---|---|---|
| D-PSGD | $(-, 0.0, 0.01, 1.0)$ | $(-, 0.0, 0.01, 1.0)$ |
| *NGC* $\alpha = 0$ (ours) | $(0.0, 0.0, 0.01, 1.0)$ | $(0.0, 0.0, 0.01, 1.0)$ |
| CGA | $(-, 0.9, 0.01, 0.5)$ | $(-, 0.9, 0.01, 0.1)$ |
| *NGC* | $(1.0, 0.9, 0.01, 0.5)$ | $(1.0, 0.9, 0.01, 0.1)$ |
| CompCGA | $(-, 0.9, 0.01, 0.1)$ | $(-, 0.9, 0.01, 0.1)$ |
| *CompNGC* | $(1.0, 0.9, 0.01, 0.1)$ | $(1.0, 0.9, 0.01, 0.1)$ |

**Hyper-parameters used for Table. 4:** All the experiments in Table. 4 have stopping criteria set to 3 epochs. We decay the learning rate by $10\times$ at $2^{nd}$ epoch. Table 10 presents the $\alpha$, $\beta$, $\eta$, and $\gamma$ corresponding to the ngc mixing weight, momentum, learning-rate and gossip averaging rate. For all the experiments, we use a mini-batch size of 32 per agent on AGNews dataset.

**Hyper-parameters for Figures:** The simulations for all the figures are run for 300 epochs. We scale the learning-rate by a factor or $0.981$ after each epoch to obtain a smoother curve. All the experiment in Figure 1, 2, 3, 4 and 5 use 5 layer CNN network. Experiments in Figure 1, and 3 use 5 agents while the experiments in Figure 2, 4 and 5 use 10 agents. The hyper-parameters for the simulations for all the plots are mentioned in Table 11

Table 10: Hyper-parameters used for Table. 4

| | BERT$_{mini}$ | | DistilBERT$_{base}$ |
|---|---|---|---|
| Method | Agents = 4 $(\alpha, \beta, \eta, \gamma)$ | Agents = 8 $(\alpha, \beta, \eta, \gamma)$ | Agents = 4 $(\alpha, \beta, \eta, \gamma)$ |
| D-PSGD | $(-, 0.0, 0.01, 1.0)$ | $(-, 0.0, 0.01, 1.0)$ | $(-, 0.0, 0.01, 1.0)$ |
| *NGC* $\alpha = 0$ (ours) | $(0.0, 0.0, 0.01, 1.0)$ | $(0.0, 0.0, 0.01, 1.0)$ | $(0.0, 0.0, 0.01, 1.0)$ |
| CGA | $(-, 0.9, 0.01, 0.5)$ | $(-, 0.9, 0.01, 0.5)$ | $(-, 0.9, 0.01, 0.5)$ |
| *NGC* | $(1.0, 0.9, 0.01, 0.5)$ | $(1.0, 0.9, 0.01, 0.5)$ | $(1.0, 0.9, 0.01, 0.5)$ |
| CompCGA | $(-, 0.9, 0.01, 0.5)$ | $(-, 0.9, 0.01, 0.5)$ | $(-, 0.9, 0.01, 0.5)$ |
| *CompNGC* | $(1.0, 0.9, 0.01, 0.5)$ | $(1.0, 0.9, 0.01, 0.5)$ | $(1.0, 0.9, 0.01, 0.5)$ |

Table 11: Hyper-parameters used for Figures 1, 2, 3, 4 and 5.

| Figure | $\alpha$ | $\beta$ | $\eta$ | $\gamma$ |
|---|---|---|---|---|
| 1a (skew=0, *NGC*) | 1.0 | 0.9 | 0.01 | 1.0 |
| 1b (skew=1, *NGC*) | 1.0 | 0.9 | 0.01 | 0.1 |
| 1c and 3 (skew=1, ring topology) | 1.0 | 0.9 | 0.01 | 0.1 |
| 2 (skew=1, *NGC*) ring topology | 0.0 | | | 1.0 |
| | 0.25 | | | 1.0 |
| | 0.5 | 0.9 | 0.01 | 0.5 |
| | 0.75 | | | 0.5 |
| | 1.0 | | | 0.25 |
| 4a (skew=0, *NGC*) | 1.0 | 0.9 | 0.1 | 1.0 |
| 4b (skew=1, *NGC*) | 1.0 | 0.9 | 0.1 | 0.5 |
| 4c (skew=1, ring topology) | 1.0 | 0.9 | 0.1 | 0.5 |

## A.5 ANALYSIS FOR 10 AGENTS

We show the convergence characteristics of the proposed *NGC* algorithm over IID and Non-IID data sampled from CIFAR-10 dataset in Figure. 4a, and 4b respectively. For Non-IID distribution, we observe that there is a slight difference in convergence rate (as expected) with slower rate for sparser topology (undirected ring graph) compared to its denser counterpart (fully connected graph). Figure. 4c shows the comparison of the convergence characteristics of the *NGC* technique with the current state-of-the-art CGA algorithm. We observe that *NGC* has lower validation loss than CGA for same decentralized setup indicating its superior performance over CGA. We also plot the model variance and data variance bias terms for both *NGC* and CGA techniques as shown in Figure. 5a, and 5b respectively. We observe that both model variance and data variance bias for *NGC* are significantly lower than CGA.

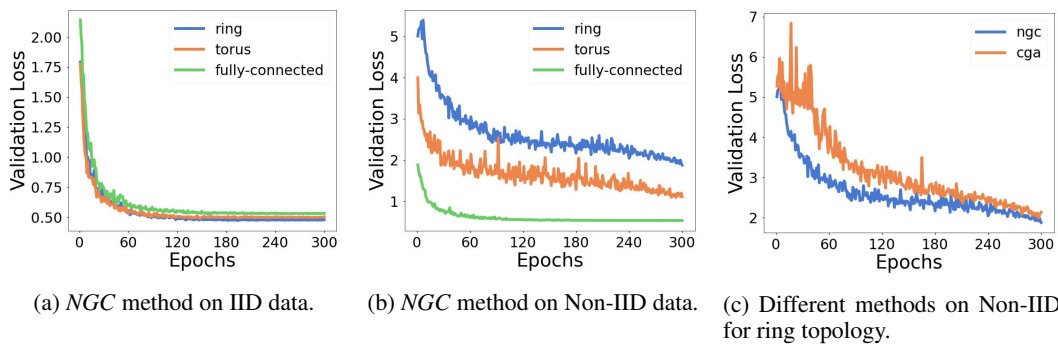

(a) *NGC* method on IID data.  (b) *NGC* method on Non-IID data.  (c) Different methods on Non-IID for ring topology.

Figure 4: Average Validation loss during training of 10 agents on CIFAR-10 dataset with a 5 layer CNN network.

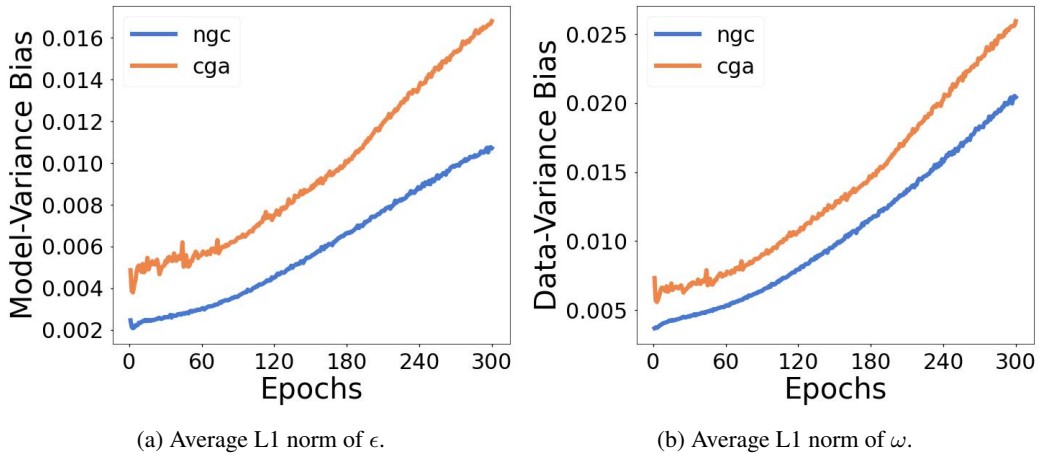

(a) Average L1 norm of $\epsilon$.

(b) Average L1 norm of $\omega$.

Figure 5: Average L1 norm of model variance bias and data variance bias for 10 agents trained on CIFAR-10 dataset with 5 layer CNN architecture over an undirected ring topology.

### A.6  COMMUNICATION COST:

In this section we present the communication cost per agent in terms of Gigabytes of data transferred during the entire training process (refer Tables. 12, 13, 15, 14). The D-PSGD and *NGC* with $\alpha = 0$ have the lowest communication cost ($1\times$). We emphasize that *NGC* with $\alpha = 0$ outperforms D-PSGD in decentralized learning over label-wise non-IID data for same communication cost. *NGC* and CGA have $2\times$ communication overhead compared to D-PSGD where as *CompNGC* and CompCGA have $1.03\times$ communication overhead compared to D-PSGD. The compressed version of *NGC* and CGA compresses the second round of cross-gradient communication to 1 bit. We assume the full-precision cross-gradients to be of 32-bit precision and hence the *CompNGC* reduces the communication cost by $32\times$ compared to *NGC*.

Table 12: Communication costs per agent in GBs for experiments in Table 1

| Method | Agents | 5layer CNN Ring | 5layer CNN Torus | VGG-11 Ring | ResNet-20 Ring |
|---|---|---|---|---|---|
| D-PSGD | 5 | 17.75 | - | 270.64 | 127.19 |
| and | 10 | 8.92 | 13.38 | 135.86 | 63.84 |
| *NGC* $\alpha = 0$ | 20 | 4.50 | 68.48 | 32.18 | |
| CGA | 5 | 35.48 | - | 541.05 | 254.27 |
| and | 10 | 17.81 | 26.72 | 271.50 | 127.59 |
| *NGC* | 20 | 8.98 | 17.95 | 136.72 | 64.25 |
| CompCGA | 5 | 18.31 | - | 279.09 | 131.16 |
| and | 10 | 9.20 | 13.79 | 140.10 | 65.84 |
| *CompNGC* | 20 | 4.64 | 9.28 | 70.61 | 33.18 |

### A.7  RESOURCE COMPARISON

The communication cost, memory overhead and compute overhead for various decentralized algorithms are shown in Table. 5. The D-PSGD algorithm requires each agent to communicate model parameters of size $m_s$ with all the $N_i$ neighbors for the gossip averaging step and hence has a communication cost of $\mathcal{O}(m_s N_i)$. In case of *NGC* and CGA, there is an additional communication round for sharing data-variant cross gradients apart from sharing model parameters for gossip averaging step. So, both these techniques incur a communication cost of $\mathcal{O}(2m_s N_i)$ and therefor an overhead of $\mathcal{O}(m_s N_i)$ compared to D-PSGD. *CompNGC* compresses the additional round of communication involved with *NGC* from $b$ bits to 1 bit. This reduces the communication overhead from $\mathcal{O}(m_s N_i)$ to $\mathcal{O}(\frac{m_s N_i}{b})$.

Table 13: Communication costs per agent in GBs for experiments in Table 2

| Method | Agents | Fashion MNIST (LeNet-5) | CIFAR-100 (ResNet-20) | Imagenette (MobileNet-V2) |
|---|---|---|---|---|
| D-PSGD and | 5 | 17.25 | 103.74 | 103.12 |
| *NGC* $\alpha = 0$ | 10 | 8.61 | 51.89 | 51.60 |
| CGA and | 5 | 34.50 | 207.47 | 206.23 |
| *NGC* (ours) | 10 | 17.23 | 103.79 | 103.19 |
| CompCGA and | 5 | 17.79 | 106.98 | 106.34 |
| *CompNGC* (ours) | 10 | 8.88 | 53.52 | 53.21 |

Table 14: Communication costs per agent in GBs for experiments in Table 3

| Method | Ring topology | chain topology |
|---|---|---|
| D-PSGD and *NGC* $\alpha = 0$ | 501.98 | 401.59 |
| CGA and *NGC* | 1003.96 | 803.17 |
| CompCGA and *CompNGC* | 517.67 | 414.14 |

CGA algorithm stores all the received data-variant cross-gradients in the form of a matrix for quadratic projection step. Hence, CGA has a memory overhead of $\mathcal{O}(m_s N_i)$ compared to D-PSGD. *NGC* does not require any additional memory as it averages the received data-variant cross-gradients into self-gradient buffer. The compressed version of *NGC* requires an additional memory of $\mathcal{O}(m_s N_i)$ to store the error variables $e_{ji}$ (refer Algorith. 2). CompCGA also needs to store error variables along with the projection matrix of compressed gradients. Therefore, CompCGA has a memory overhead of $\mathcal{O}(m_s N_i + \frac{m_s N_i}{b})$. Note that memory overhead depends on the type of graph topology and model architecture but not on the size of the graph. The memory overhead for different model architectures trained on undirected ring topology is shown in Table. 16

The computation of the cross-gradients (in both CGA and NGC algorithms) requires $N_i$ forward and backward passes through the deep learning model at each agent. This is reflected as $\mathcal{O}(3N_i FP)$ in the compute overhead in Table. 5. We assume that the compute effort required for the backward pass is twice that of the forward pass . CGA algorithm involves quadratic programming projection step Goldfarb & Idnani (1983) to update the local gradients. Quadratic programming solver (quadprog) uses Goldfarb/Idnani dual algorithm. CGA uses quadratic programming to solve the following (Equation 6 -see Equation 5a in Esfandiari et al. (2021)) optimization problem in an iterative manner:

$$\min_u \quad \frac{1}{2} u^T G G^T u + g^T G^T u$$
$$\text{s.t.} \quad u \geq 0 \tag{6}$$

Table 15: Communication costs per agent in GBs for experiments in Table 4

| Method | BERT$_{\text{mini}}$ | | DistilBERT$_{\text{base}}$ |
|---|---|---|---|
| | Agents = 4 | Agents = 8 | Agents = 4 |
| D-PSGD and *NGC* $\alpha = 0$ | 234.30 | 118.20 | 1410.39 |
| CGA and *NGC* | 486.59 | 236.40 | 2820.77 |
| CompCGA and *CompNGC* | 241.6 | 121.89 | 1454.46 |

Table 16: Memory overheads for various methods trained on different model architectures with CIFAR-10 dataset over undirected ring topology with 2 neighbors per agent.

| Architecture | CGA (MB) | NGC (MB) | CompCGA (MB) | CompNGC (MB) |
|---|---|---|---|---|
| 5 layer CNN | 0.58 | 0 | 0.58 | 0.60 |
| VGG-11 | 4.42 | 0 | 4.42 | 4.56 |
| ResNet-20 | 2.28 | 0 | 2.28 | 2.15 |

where, G is the matrix containing cross-gradients, g is the self-gradient and the optimal gradient direction $g^*$ in terms of the optimal solution of the above equation $u^*$ is $g^* = G^T u^* + g$. The above optimization takes multiple iterations which results in compute and time complexity to be of polynomial(degree$\geq$ 2) order. In contrast, NGC involves simple averaging step that requires $O(m_s N_i)$ addition operations.

