# OpenReview forum: "Neighborhood Gradient Clustering: An Efficient Decentralized Learning Method for Non-IID Data Distributions"
_ICLR.cc/2023/Conference — Submitted to ICLR 2023_

### Official Review · Reviewer_b2jc · 2022-10-22

**Confidence:** 5
**Correctness:** 3
**Technical Novelty And Significance:** 1
**Empirical Novelty And Significance:** 1
**Recommendation:** 5

**Clarity, Quality, Novelty And Reproducibility:**

The clarity and quality of the presentation in this paper look good, but the big weakness as mentioned above is the novelty, which is quite marginal. The current form of the paper cannot provide sufficient contributions.

**Strength And Weaknesses:**

Strength: this paper presents a novel algorithm on top of an existing one and shows the better model performance compared to the existing baseline. The investigated topic is quite interesting as non-IID remains a challenging problem in the decentralized learning area. The paper is easy to follow and the presentation is clear.

Weaknesses: the novelty in this work is quite marginal. It looks the only novelty is to replace the gradient calculation step in CGA with the weighted mean of self-gradients, model-variant cross-gradients, and data-variant cross-gradients. Such a change is just a minor incremental work, which fails to provide sufficient contributions. Also, concepts are just based on the existing works. Given this, if the authors could provide detailed theoretical analysis on NGC and CompNGC, that may make this paper technically solid and sound. However, the authors only provide empirical results, which to me is not promising as well. If this paper is posited as an applied paper, then comprehensive results are necessarily required, including diverse model architectures, datasets, tasks, topologies and numbers of agents. Through these, if NGC and CompNGC remain competitive and outperform baselines, then the work will look very technically solid and sound. However, the existing experimental results to me are not very comprehensive and promising.

*******************************Post-rebuttal************************************
I appreciate the detailed response and additional results in the revised paper from the authors. After carefully reviewing the responses, I think the additional results have made the work better. However, the overall novelties in this work is still low to me and it still has much room for improvement in terms of theory. Though I would raise my score given the much better empirical evidences for the proposed algorithm in the revised version, the paper with the current form is still marginally below the acceptance standard.

**Summary Of The Paper:**

This paper presents a new decentralized learning algorithm that bases on the proposed cross gradient aggregation (CGA) algorithm. Specifically, the authors leverage the existing self-gradient and cross-gradient concepts to develop the neighborhood gradient clustering (NGC) algorithm. This proposed method replaces the local gradients of the model with the weighted mean of the self-gradients, model-variant cross-gradients, and data-variant cross-gradients, which is better able to handle the parameter variations in non-IID scenarios. To reduce the communication bottleneck, the authors develop the compressed version of NGC, CompNGC. To validate both algorithms, the authors utilize a benchmark dataset with various model architectures and the results show the superiority of the proposed algorithms.

**Summary Of The Review:**

This paper presents a new decentralized learning algorithm called NGC to cope with the issues in decentralized learning settings. The authors provide algorithm design and empirical evidence to show case the superiority. This work seems incremental, failing to providing good novelties. The authors should analyze the proposed algorithms theoretically and present more experimental result to validate.

---

### Official Review · Reviewer_LFjx · 2022-10-25

**Confidence:** 4
**Clarity, Quality, Novelty And Reproducibility:** Look good to me.
**Correctness:** 3
**Technical Novelty And Significance:** 2
**Empirical Novelty And Significance:** 2
**Recommendation:** 5

**Strength And Weaknesses:**

Strength
1. The motivation of this paper is clear and interesting.
2. The proposed method looks reasonable.
3. Experiments have some promising results.

Weaknesses
1. As pointed out in Introduction section, the idea of leveraging cross-gradient information has already been used in Esfandiari et al. (2021). Therefore, the novelty of this paper is limited in this sense.
2. It would be great to include some convergence analysis in this paper. Many decentralized learning methods have convergence analysis, such as D-PSGD. Actually, Esfandiari et al. (2021) which also uses cross-gradient information, has convergence analysis as well.
3. The experiments are only conducted on one dataset, i.e., CIFAR10. It’s better to use more datasets. Decentralized learning algorithms are particularly useful for training large models on large distributed datasets. The authors are encouraged to use large datasets (such as ImageNet).


**Summary Of The Paper:**

This paper studies the decentralized distributed training under the non-iid data distribution setting. Different from the baseline method D-PSGD, it leverages self- and cross-gradient information to modify the gradient update for each agent.

**Summary Of The Review:**

The paper proposed an interesting method for decentralized learning but it has several weaknesses.

---

### Official Review · Reviewer_y4qJ · 2022-10-25

**Confidence:** 4
**Correctness:** 3
**Technical Novelty And Significance:** 3
**Empirical Novelty And Significance:** 2
**Recommendation:** 5

**Clarity, Quality, Novelty And Reproducibility:**

The presentation is very clear and easy to follow. The proposed methodology is novel but needs a more comprehensive justification.

**Strength And Weaknesses:**

Strength: the paper is quite well-written and easy to follow.
Weakness: the working mechanism of the proposed method is not fully demonstrated. Other than the empirical performance on a benchmark data set, it is not clear to me why this method work or under what circumstances the proposed method will work best.

1. The paper has emphasized the concept of "non-IID" data. But what types of data can this method handle? It would be very ambitious to claim that it will work for any type of setting. So it is necessary to clarify or at least give some concrete examples of the non-IID data types under consideration.

2. What types of models are most suited for the proposed method? Regression problems? Classification problems? or other tasks? Different models have very different structures on the gradients and one needs to be more specific.

3. The term "clustering" in the title implies some sort of unsupervised learning algorithms which I do not see in the paper.  To the best of my understanding, the paper seems to assume that the data distributions among the neighboring nodes are somehow "homogeneous". Otherwise, I do not understand why taking a weighted average of a bunch of inhomogeneous gradients will help improve the model's performance.  Please clarify.

4. Since there are no theoretical justifications for the proposed method, I think carefully designed synthetic experiments are necessary to study the advantages and limitations of the proposed method. Otherwise, I have little confidence in the generalizability of the method.


**Summary Of The Paper:**

This paper proposed a new algorithm that utilizes groups of stochastic gradients collected from neighborhood nodes in a decentralized learning environment with heterogeneous data distributed in different locations. The proposed algorithm is communication efficient and achieves better empirical performance on non-IID data sampled from the CIFAR-10 Dataset.

**Summary Of The Review:**

The paper is very well written but the supporting evidence for the model performance is not sufficient.

---

### Author Response · Authors · 2022-11-17
**Summary of the final additions to the rebuttal (revised paper)**

The following are the changes made to the revised version of the paper. We included the highlighted paper with the changes marked in blue in the supplementary folder.
* Updated the contributions to clarify the novelty of the proposed algorithm
* Updated the experimental section and appendix with an exhaustive set of experiments. All the experiments were conducted for three different seeds and the averaged test accuracy and the standard deviation are reported for statistical significance.
  * CIFAR10 on various architectures such as 5layer CNN, VGG11, and Resnet-20 over 5, 10, and 20 agents with ring and torus topologies (existing)
  * CIFAR100 trained on ResNet-20 architecture over 5 and 10 agents ring topology
  * Fashion MNIST trained on LeNet-5 architecture over 5 and 10 agents ring topology
  * Imagenette (low resolution - 32x32) trained on MobileNet-V2 over 5 and 10 agents ring topology.
    * Imagenette is a 10-class subset of the ImageNet dataset.
  * Imagenette (high resolution - 224x224) trained on ResNet-18 over 5 agents with ring and chain topology
  * AGNews (text classification) trained on BERT$_{mini}$ over 4 and 8 agents ring topology
  * AGNews trained on DistilBERT trained on 4 agents ring topology

---

### Decision · Program_Chairs · 2023-01-20

**Decision:**

Reject

**Justification For Why Not Higher Score:**

NA

**Justification For Why Not Lower Score:**

NA

**Metareview: Summary, Strengths And Weaknesses:**

This work considers decentralized optimization problems where agents' local data sets are non-i.i.d. To mitigate this data heterogeneity issue, an augmentation of a pre-existing decentralized optimization method is developed based on local weighted averaging of agents' gradients, which is termed (somewhat erroneously) as gradient clustering. This method bears a number of resemblances to cross gradients as detailed in the following reference:

[1] Esfandiari, Yasaman, et al. "Cross-gradient aggregation for decentralized learning from non-iid data." International Conference on Machine Learning. PMLR, 2021.

and does not provide any specific theoretical guarantees of the proposed modification on top of prior work. Therefore, the technical novelty and innovation is not very clear. However, experimental evidence demonstrates the merits of the proposed approach in a limited manner. During the rebuttal phase, the authors have significantly expanded the degree of experimental validation, which brings the bar closer to meeting the standard of acceptance. However, without sound conceptual underpinning and an explicit relationship between the proposed methodology and the actual degree of data heterogeneity as quantified by some bias or power law effect of the data distribution, it is unclear what is the technical contribution. For this reason, the work is not yet warranting acceptance.